# Sarcopenic Obesity in People with Alcoholic Use Disorder: Relation with Inflammation, Vascular Risk Factors and Serum Vitamin D Levels

**DOI:** 10.3390/ijms24129976

**Published:** 2023-06-09

**Authors:** Candelaria Martín-González, Paula Fernández-Alonso, Onán Pérez-Hernández, Pedro Abreu-González, Elisa Espelosín-Ortega, Camino María Fernández-Rodríguez, Esther Martín-Ponce, Emilio González-Reimers

**Affiliations:** 1Departamento de Medicina Interna, Universidad de La Laguna, Servicio de Medicina Interna, Hospital Universitario de Canarias, Tenerife, Canary Islands, 38320 La Laguna, Spain; paulafdezalonso@gmail.com (P.F.-A.); onanperezhernandez@gmail.com (O.P.-H.); caminoferro@gmail.com (C.M.F.-R.); esthermartinp@yahoo.es (E.M.-P.); egonrey@ull.edu.es (E.G.-R.); 2Departamento de Ciencias Médicas Básicas, Unidad de Fisiología, Universidad de la Laguna, Tenerife, Canary Islands, 38320 La Laguna, Spain; pabreu@ull.edu.es; 3Servicio de Laboratorio, Hospital Universitario de Canarias, Tenerife, Canary Islands, 38320 La Laguna, Spain; elisaespelosin@gmail.com

**Keywords:** alcoholism, sarcopenic obesity, handgrip, vitamin D, proinflammatory cytokines

## Abstract

In recent years, the terms sarcopenia, sarcopenic obesity, and osteosarcopenic obesity (OSO) were coined to define a situation in elderly people strongly associated with frailty and increased mortality. Possibly, a complex interplay of several hormones and cytokines are involved in its development. Ongoing research detected that OSO may occur at any age and in several conditions. The prevalence of OSO in alcoholism was poorly analyzed. The aim of this study was to analyze the prevalence of OSO in alcoholism and its relationship with proinflammatory cytokines and/or common complications of alcoholism, such as cirrhosis, cancer, or vascular disease. We included 115 patients with alcoholic use disorder. Body composition analysis was performed by double X-ray absorptiometry. Handgrip strength was recorded using a dynamometer. We assessed liver function according to Child’s classification, and determined serum levels of proinflammatory cytokines (TNF-α, IL-6, IL-8), routine laboratory variables, and vitamin D. People with alcoholic use disorder showed a high prevalence of OSO, especially regarding OSO obesity (60%), OSO osteopenia (55.65%), and OSO lean mass (60.17%). OSO handgrip was closely, independently, related to the presence of vascular calcification (χ^2^ = 17.00; *p* < 0.001). OSO handgrip was related to several proinflammatory cytokines and vitamin D. Vitamin D deficiency kept a close correlation with OSO handgrip (rho = −0.54, *p* < 0.001). Therefore, among people with alcohol use disorder, OSO prevalence was high. OSO handgrip is related to serum proinflammatory cytokine levels supporting the possible pathogenetic role of these cytokines on OSO development. Vitamin D deficiency is related to OSO handgrip suggesting its pathogenetic involvement in sarcopenia in patients with alcohol use disorder. The close association between OSO handgrip and vascular calcification is clinically relevant and suggests that OSO handgrip may constitute a prognostic tool in these patients.

## 1. Introduction

In recent years, the terms sarcopenia, sarcopenic obesity, and osteosarcopenic obesity (OSO) were coined to define a situation in elderly people strongly associated with frailty [1] and increased mortality. Initially, OSO was considered a physiologic condition in the elderly. For instance, some authors reported that muscle mass decreases normally at an estimated rate of 4% per decade after the age of 50 years [2], whereas other authors reported a loss of muscle mass at a rate of 1–2% per year, with an even more marked decrease in muscle strength (1.5–3% per year) after the fourth decade. These muscle alterations are accompanied by a similar decrease in bone mass which is more intense among women around the menopause [3]. Age-associated adiposity was interpreted as secondary to decreased activity related to sarcopenia. However, ongoing research detected that OSO may occur at any age and in several conditions. Currently, OSO should be considered a true illness with a high prevalence in the elderly (but also affecting young people), a complex pathophysiology and a poor prognosis [4], rather than a mere association of alterations in body composition related to the ageing process. Reported prevalence rates in middle-aged and/or elderly people vary according to geographical area and diagnostic criteria. For instance, in a Korean population-based study including 5918 individuals aged > 50 years, 73.73% of men and 89.8% of women showed at least one OSO criterion [5]. In America, sarcopenia affects roughly 50% of adults > 80 years old [6]. In a French study including 1409 healthy individuals aged > 45 years, the prevalence of sarcopenia was 16.1%, and 3.4% of women and 3.5% of men fulfilled all three of the OSO criteria [7]. In a recent meta-analysis, the prevalence of OSO in middle-aged and older adults worldwide was 8%, increasing to 13% in the elderly [8], but OSO criteria (increased adiposity with reduced muscle and bone mass) were also observed among obese individuals aged 18–21 years [9]. Therefore, OSO may also affect apparently healthy young individuals.

Mechanisms leading to OSO are only partially known, but it is currently clear that increased fat deposition is by no means only a mere consequence of altered intake and/or decreased activity induced by sarcopenia, but also a source of proinflammatory cytokines [10] that contribute to the development of the full-blown OSO syndrome, leading to muscle atrophy, secondary osteopenia/osteoporosis and, probably, to systemic manifestations involving remote organs, as, for instance, brain dysfunction [11] or vascular disease or hypertension [12], that ultimately cause increased mortality.

The definition of OSO is based on the presence of reduced bone mass, muscle atrophy, and increased body fat. Normal-nourished individuals show a distribution of fat mass, muscle mass, and bone mass within the normal limits, usually defined by densitometric or anthropometric criteria [13,14], or by subjective nutritional scores that are associated with the lowest mortality and morbidity rates [15]. Proteins, carbohydrates, and fats are essential components of body composition, and there is a complex interplay among these three kinds of molecules. Proteins may be considered a structural component of muscle and bone matrix, but also a source of amino acids that may be preferentially used by the liver to synthesize glucose, albumin, coagulation factors and acute phase reactants, and, in a small proportion, directly as fuel [16]. On the other hand, fat and carbohydrates are main fuel reserves, providing calories for the function of the metabolic pathways necessary to live. Adequate dietary intake of fat, carbohydrates, and proteins is essential, but if the demand for acute phase reactants and/or metabolic activity increases, biochemical changes take place [17]. For instance, liver synthesis of circulating proteins becomes a priority over the maintenance of adequate muscle mass; and, if intake is poor, fat reserve is utilized to maintain an adequate caloric supply. Therefore, nutritional status not only depends on food availability, but also on the ability of the body to respond to a modification of food availability and/or metabolic requirements. Complex hormonal and cytokine networks govern the metabolic changes that take place when starvation and/or metabolic stress ensue. The delicate equilibrium between the tripod formed by muscle mass, fat deposits, and bone mass may be distorted when a superimposed factor disrupts this hormonal and cytokine network, and/or impairs the structure of the tripod, directly altering fat, bone, or muscle mass by a more or less selective toxic effect. This is the case of ethanol, whose consumption is associated with increased expression of proinflammatory cytokines [18], several hormonal changes, and a direct toxic effect on muscle, among other actions. In addition, people with severe alcohol use disorder often develop a bizarre style of life, losing interest for food, altering eating behavior [19] and frequently develop an alteration of gastric emptying [20], pancreatic damage [21], or liver cirrhosis with portal hypertension. These entities may aggravate malabsorption, also directly induced by ethanol consumption. Moreover, infections may be more frequent and severe in people with alcohol use disorder [22], leading to frequent episodes of the so-called kwashiorkor-like malnutrition, with severe muscle wasting. All these alterations are even more complex, since adipose tissue, muscle and bone are also sources of adipokines, myokines, and osteokines, respectively, that participate in the metabolic response [23,24]. Poor intake and/or malabsorption may alter vitamin D levels, a hormone with important effects on bone, fat and muscle, although its pathogenetic role in OSO is a matter subjected to debate [25]. In addition, as commented, direct toxic effects on muscle and bone, described in people with alcohol use disorder since a long time ago, and aggravated in patients with cirrhosis [26], may also add to the aforementioned factors. For these reasons, the prevalence of OSO should be high among severe people with alcohol use disorder, and likely, a relationship should exist among proinflammatory status and/or altered hormonal levels and OSO. Since OSO may be related to several complications and a poor prognosis at least in the elderly, the assessment of its prevalence among alcoholic patients is especially relevant, because heavy drinkers usually develop organic complications (affecting muscles, bone, nutritional status, among others) at earlier ages. Therefore, identification of a potentially reversible condition such as OSO in these patients is of paramount importance, as is the detection of a possible relationship between OSO and the proinflammatory status and/or altered hormonal profile associated with alcoholism.

Based on these facts, the aim of the present study is to analyze the prevalence of OSO criteria (obesity, handgrip/lean mass, osteopenia/osteoporosis) among people with alcohol use disorder, the relationship of OSO with some common complications of alcoholism, such as cirrhosis, hypertension [27], cancer [28], or vascular calcifications [29], and with proinflammatory cytokine and vitamin D levels, given the possible pathogenetic role of inflammation on OSO development, and the relevance of the metabolic role of vitamin D in bone, muscle, and adipose tissue.

## 2. Results

The prevalence of OSO osteopenia reached 55.65%; that of OSO osteoporosis reached 19.13%; that of OSO obesity, 60%; that of OSO handgrip, 33.91%, and that of OSO lean mass, 60.17%. Among people without alcohol use disorder, the prevalence of OSO handgrip was significantly lower (8.82%; χ^2^ = 6.983; *p* = 0.008). A similar difference was observed regarding OSO osteopenia (30.8%; χ^2^ = 4.31; *p* = 0.038), but not regarding OSO osteoporosis, which was present in 7.7% (χ^2^ = 1.96; *p* = 0.25)

The prevalence of each of the OSO criteria among people with cirrhosis and without cirrhosis is shown in Table 1. Overall, each OSO criteria (other than OSO obesity) showed a trend towards a higher prevalence among people with cirrhosis, although these associations were not statistically significant.

Additionally, people with cirrhosis showed a non-significant trend to present more OSO criteria than people without cirrhosis (16.66% of those with the condition showed four or five OSO criteria, vs. 10.17% of those without it; χ^2^ = 1.23; *p* = 0.27).

### 2.1. Liver Function and Amount of Ethanol Consumption

#### 2.1.1. OSO Osteoporosis/Osteopenia

OSO osteoporosis was related to the daily amount of ethanol consumed (Z = 2.16; *p* = 0.03); those with the OSO criterion consumed higher amounts. Serum albumin was lower in patients with OSO osteoporosis (3.10 ± 0.79 g/dL) compared with levels among patients without osteoporosis (3.42 ± 0.67 g/dL) (Z = 1.99; *p* = 0.047). Prothrombin activity was also lower among patients with OSO osteoporosis (68.69% ± 22.58%) than in those without OSO osteoporosis (78.90% ± 20.94%, Z = 2.06; *p* = 0.039), but no differences were observed regarding age or serum bilirubin. No relationships were observed between OSO osteopenia and any of the mentioned variables.

#### 2.1.2. OSO Obesity

It was related to age (t = 2.90; *p* = 0.004), being higher in patients with OSO obesity (57.71 ± 12.72 years) compared with levels among patients without OSO obesity (58.54 ± 13.24 vs. 50.93 ± 9.70 years, t = 3.55; *p* = 0.001), but no differences were observed with prothrombin, serum albumin or bilirubin, daily ethanol consumption, or years of addiction.

#### 2.1.3. OSO Handgrip

It was related to age (t = 3.58; *p* = 0.001), which was higher in patients with OSO handgrip (61.03 ± 11.08 years) compared with patients without OSO handgrip (52.66 ± 12.26 years), but no differences were observed with prothrombin (Z = 1.16) or serum albumin (Z = 0.41). Patients with OSO handgrip showed higher bilirubin (Z = 2.70; *p* = 0.007) and platelet count (Z = 2.14; *p* = 0.032). OSO handgrip was related to years of addiction (Z = 2.39; *p* = 0.017), but this relationship was displaced by age when a multivariate analysis was performed.

#### 2.1.4. OSO Lean Mass

It was related to age (t = 3.09; *p* = 0.002), being higher in patients with the OSO lean mass criterion (57.87 ± 12.07 years) compared with those without OSO lean mass criterion (51.86 ± 11.02 years). It was also related to serum albumin (Z = 2.70; *p* = 0.007), being lower among patients with OSO lean mass criterion (3.26 ± 0.66 g/dL vs. 3.50 ± 0.78 g/dL). Handgrip was lower in patients with the OSO lean mass criterion (Z = 2.67; *p* = 0.007) and total hip T-score was also lower in patients with the OSO lean mass criterion (Z = 1.98; *p* = 0.047).

#### 2.1.5. Overall OSO

As expected, there was a clear-cut association between OSO and age (F = 11.90; *p* < 0.001. Figure 1). Patients with four or five OSO criteria were significantly older than those with two or three criteria, or those with zero or one criteria. In relation to this result, patients with four or five OSO criteria were drinking for a longer time than those with zero or one OSO criteria, but the significant relationship with years of addiction (Z = 2.09; *p* = 0.034) was displaced by age by logistic regression analysis.

Marked differences were observed in total hip T-score, proportion of fat, and handgrip strength when these variables were compared among patients with zero to one OSO criteria, two or three OSO criteria, or four or five OSO criteria (Table 2).

In addition, prothrombin activity was significantly reduced among individuals with two or more OSO criteria than in those with zero or one OSO criteria (F = 4.38; *p* = 0.015), and vitamin D levels (KW = 6.78; *p* = 0.034. Figure 2) were significantly different among the three OSO groups.

### 2.2. Inflammatory Cytokines and Vitamin D

The patients showed altered levels of vitamin D when compared with the standard values of our laboratory (Table 3). Only 14 out of 56 patients with cirrhosis (25%) and 14 out of 59 patients without it (23.7%) showed normal vitamin D values. Regarding inflammatory cytokines, IL-6 and IL-8 values were clearly higher than those observed in healthy, hospital workers, drinkers of less than 10 g ethanol/day, but no differences were observed regarding serum TNF-α (Table 3.)

#### OSO Handgrip

Marked differences were observed among patients with or without OSO handgrip criterion regarding TNF-α (Z = 3.67; *p* < 0.001), IL-6 (Z = 2.68; *p* = 0.007), IL-8 (Z = 3.61; *p* < 0.001), and vitamin D (Z = 5.25; *p* < 0.001). Indeed, significant relationships were observed between IL-8 and handgrip (rho = −0.28; *p* = 0.003), between IL-6 and handgrip (rho = −0.38; *p* < 0.001), and between handgrip and vitamin D (rho = −0.54, *p* < 0.001), but the relationship between handgrip and TNF-α was a direct one (rho = 0.32; *p* < 0.001). A multiple correlation analysis including age, proinflammatory cytokines, and vitamin D showed that vitamin D (directly), IL-6 (inversely), and age (inversely), in this order, were all independently related to handgrip (Table 4).

When multiple correlation analyses were performed including only two independent variables (age and IL-6, age and IL-8, age and vitamin D, age and TNF-α), most relationships among handgrip and inflammatory cytokines were displaced by age, but the relationship with vitamin D was independent of age, entering first place in the stepwise multiple correlation analysis.

After classification of IL-6, IL-8, vitamin D, TNF-α, and age as dichotomic variables according to medians to perform a logistic regression analysis, we found that vitamin D, age, and IL-8 were independently related to OSO handgrip. Vitamin D was the first variable selected, showing higher values in patients without OSO handgrip criterion, in contrast with IL-8 and age, who showed higher values in patients with OSO handgrip criterion (Table 5).

Patients with OSO handgrip showed an increased amount of trunk fat (11673 ± 6010 g vs. 9701 ± 5527 g; Z = 1.98; *p* = 0.047). An inverse correlation was observed between handgrip and trunk fat mass (rho = −0.21; *p* = 0.027), although not with total fat (rho = 0.07).

### 2.3. Relationships with Hypertension, Diabetes, and Vascular Calcifications

No associations were observed among any OSO criteria and the presence of diabetes (25 patients), hypertension (30 patients), or cancer (30 patients), but OSO handgrip was significantly associated with vascular calcification (χ^2^ = 17.00; *p* < 0.001. Figure 3), even though when the nine patients with serum creatinine ≥ 1.40 mg/dL were excluded (χ^2^ = 17.90; *p* < 0.001).

Vitamin D was also significantly lower among patients with vascular calcification (t = 2.44; *p* = 0.016), as well as older age (t = 4.04; *p* < 0.001). However, performing a logistic regression analysis between the presence of calcifications and OSO handgrip, age, and vitamin D (classified as dichotomic variables according to median values), OSO handgrip was the first variable selected (before age), and vitamin D was displaced (Table 6).

No associations were observed when the remaining OSO criteria were compared with the presence of vascular calcification. However, a trend (linear by linear association = 5.11; *p* = 0.024) was observed when vascular calcifications were compared among patients classified into three groups according to the number of OSO criteria (0, 1; 2, 3; 4, 5).

## 3. Discussion

The aims of this study were to analyze the prevalence of OSO among people with alcohol use disorder, and the relationship of OSO with some common complications of people with alcohol use disorder, such as hypertension, cancer, diabetes, or vascular calcifications. Considering the possible pathogenetic role of inflammation on OSO development, we also aimed to analyze the relationship between proinflammatory cytokine levels and OSO in people with alcohol use disorder. Given the well-described alteration of vitamin D in people with alcohol use disorder, the relevance of the metabolic role of vitamin D both on bone and muscle, and its strong, inverse association with obesity [30], we also included the analysis of the behavior of this hormone in relation to OSO.

There are three findings in this study which deserve comments. At first, the prevalence of OSO among people with alcohol use disorder is high, especially with respect to OSO obesity, OSO osteopenia, and OSO lean mass. Secondly, the clinical relevance of the presence of OSO, particularly regarding OSO handgrip, is important, because it seems to be closely related to vascular calcification in the present study, a well-known prognostic factor [31,32]. Thirdly, OSO handgrip is also related to several proinflammatory cytokines (supporting the possible pathogenetic role of these cytokines on OSO development) and vitamin D. Low vitamin D levels keep, indeed, a close correlation with OSO handgrip.

We found that OSO criteria were frequent among the population of people with alcohol use disorder: 73.5% were affected by at least two of the OSO criteria used in this study; OSO obesity, OSO osteopenia, and OSO lean mass were observed in more than 50% of the patients, and OSO handgrip in a third of them. As expected, there was a clear-cut association between OSO and age (F = 11.90; *p* < 0.001. Figure 1) and also a (non-significant) trend to a higher OSO prevalence among patients with cirrhosis.

As commented earlier, ethanol exerts many direct toxic effects on muscle [33,34], and several other mechanisms contribute to muscle wasting in people with alcohol use disorder, especially if cirrhosis ensues [35,36,37], leading to a high prevalence of sarcopenia among excessive drinkers [38]. In contrast with these observations, some authors did not find that ethanol consumption might be a risk factor for sarcopenia [39], and some data suggest that the effects of moderate ethanol consumption are minimal, provided adequate exercise is undertaken [40]. In this study, we found that over 60% of the included patients showed reduced muscle mass and over 33% showed reduced muscle strength, figures which are in accordance with most of the reported data regarding muscle damage in alcoholics [41,42,43]. Mechanistically, reduced muscle strength and muscle mass may play a key role in the development of full-blown osteosarcopenic adiposity in these patients. Less muscle mass and activity demand fewer fatty acids as fuel, a factor that may contribute to fat accumulation. In addition, alcohol inhibits fat oxidation [44], adding to the previously mentioned mechanism. Therefore, muscle wasting in alcoholics together with the direct effects of ethanol on fat metabolism could explain the high prevalence of OSO obesity. Although a great deal of controversy exists, moderate drinking may not be associated with obesity, in contrast with heavy drinkers [45], as was the case for the patients included in this study. Whatever the mechanisms involved, fat accumulation, especially trunk fat, may induce sarcopenia by itself [46]. Adipose tissue is a significant source of proinflammatory cytokines, such as IL-1β and IL-6, both probably involved in the development of sarcopenia, and TNF-α [10]. A major effect of TNF-α is the induction of insulin resistance [47], impairing protein synthesis, glucose uptake, and adequate muscle performance. However, the effects of TNF-α on muscle are more complex, since it was also described as an enhancer of glycolytic activity [48], and TNF-α is involved in muscle regeneration after injury [49]. Moreover, in obese patients, or after injury, inflammatory M-1 macrophages may infiltrate muscle tissue [50]. These macrophages secrete proinflammatory cytokines, such as TNF-α and IL-6, among others. Beenakker et al. showed that TNF-α secreted by these M-1 monocytes was significantly directly associated with lean body mass, appendicular lean mass, and handgrip strength in 191 men aged 60–70 years [51]. Perhaps a similar effect may play a role in ethanol-injured muscle and explain the direct correlation observed with handgrip strength in this study. The effect of IL-6 on muscles are also debatable [52]. Both amyotrophic and regenerative effects were described. IL-8 may also have myogenic effects [53]. Therefore, although the inverse relationship in this case with handgrip supports a pro-atrophy effect of IL-6 and IL-8, the role of proinflammatory cytokines on muscle remains uncertain. Other authors proposed that inflammatory cytokines are related to sarcopenia not with a direct effect on muscles but causing anorexia and decreasing nutrient intake [54].

In addition, sarcopenic muscle reduces mobility, therefore favoring more fat accumulation. In the present study, a striking result was the very marked difference in trunk fat when patients with OSO handgrip were compared. Trunk fat mass was significantly higher among those with the OSO handgrip criterion (Z = 1.98; *p* = 0.047), a result in accordance with what just commented. On the other hand, we observed that handgrip was closely related to inflammatory cytokines and vitamin D. Logistic regression analysis clearly showed that vitamin D and IL-6 (in addition to age) were significantly related to OSO handgrip. The importance of vitamin D in sarcopenia is fully in accordance with the well-known effects of vitamin D on muscle [55], leading to type II fiber atrophy, especially in the proximal muscles.

Strikingly, patients affected by OSO handgrip showed a marked increase in the prevalence of vascular calcifications. This is an important clinical finding since vascular calcification may be associated with increased mortality [56]. Interestingly, vascular calcification may also belong to the consequences of a smoldering pro-inflammatory milieu, eventually related to obesity [57]. Several authors reported the relationship of sarcopenia with a poor prognosis in patients with advanced liver disease [58,59]; (among others), although the definition of sarcopenia was based on diverse anatomic (radiologic) criteria [60].

OSO osteopenia was also commonly observed in our study, but OSO osteoporosis was less frequent. Among those with cirrhosis, the prevalence of osteoporosis was in the range or even higher than data reported by other authors [60]. Despite some controversy [61], there is general agreement with respect to classic observations about the deleterious effect of ethanol on bone [62], both directly [63,64] and also by inducing changes in vitamin D and other hormones when the liver failure ensues [65]. Indeed, in our study, 75% of those with cirrhosis and 76.3% of those without showed vitamin D values below 30 ng/mL, figures that are similar to those reported by others [66]. Again, reduced muscle mass and activity also contribute to decreased osteoblastic activity and bone synthesis [67]. Both the more marked hormonal changes in liver cirrhosis and the more intense sarcopenia may explain the relationship between OSO osteoporosis and variables related to liver function, such as prothrombin activity or serum albumin.

## 4. Materials and Methods

This was a retrospective analysis of prospectively collected data from cohorts of people with alcohol use disorder, destined to analyze the behavior of different variables in relation to ethanol consumption, liver function derangement, brain alterations, or bone disease. All these patients were consecutively admitted to our hospitalization unit due to organic complications related to excessive ethanol ingestion, and all of them underwent complete laboratory evaluation and body composition analysis at our hospital, which is the reference hospital for a population of ≈500,000 inhabitants, most of them living in rural areas or small towns. All the patients included in this study were attended by some of the authors during a period spanning from 2010 to 2017, who recorded all the clinical variables required, including those necessary to assess OSO criteria. Serum cytokine and vitamin D levels (required inclusion criteria) were determined at the time of inclusion, when the patients were already recovered from the acute condition that prompted hospital admission. Ten patients were women, and median (interquartile range) age of the patients was 55 (45–64) years.

For comparative purposes, we provided the median values and interquartile ranges of proinflammatory cytokines of 34 healthy hospital workers (of similar age (54 (43–58 years; Z = 1.77; *p* = 0.08) and sex (4 women; χ^2^ = 0.04; *p* = 0.84), and the reference values of 25-hydroxyvitamin D (vitamin D) (Table 3).

Patients who consumed other drugs besides tobacco were not included in the study. Alcohol consumption was assessed by direct inquiry (to the patients and/or their relatives) recording the type of beverage and the daily amount consumed, calculating (volume of ethanol consumed = degree of beverage (in %) × beverage volume) × alcohol density (0.8) to estimate consumption in g.

A total of 115 patients (10 women) were included, 56 were patients with cirrhosis, aged 56.70 ± 10.87 years, and 59 without cirrhosis, aged 54.36 ± 13.83 years (t = 1.01; NS). The diagnosis of liver cirrhosis was based on clinical data and on liver ultrasound (US) examination, considering the presence of splenomegaly and/or portal dilatation and a heterogeneous liver structure and irregular shape, together with altered levels either of albumin (<3.5 g/dL), bilirubin (1.5 mg/dL), or prothrombin activity (<75%). To achieve a global assessment of liver function, we applied the Child–Pugh score to the whole sample, despite being aware that this score was initially designed as a prognostic tool only for patients with cirrhosis. Child’s score is based on the alteration of the following variables: serum albumin, bilirubin, prothrombin activity, and presence/severity of ascites and/or encephalopathy [68,69]. Twenty-five patients also had diabetes, 30 were affected by hypertension, 30 had been or were affected by cancer at the time of inclusion, and 49 presented vascular calcification of the thoracic aorta.

The study protocol was approved by the local ethical committee of our hospital (number CHUC_2019_83) and conformed to the ethical guidelines of the 1975 Declaration of Helsinki. All the patients gave their written informed consent.

### 4.1. Body Composition

All patients underwent a total body composition analysis evaluated by densitometry, using a LUNAR PRODIGY ADVANCE device, General Electric, Piscataway, NJ, USA, following standard criteria [70] recording bone mineral density (BMD) and bone mineral content (BMC), fat and lean mass, at right arm, left arm, right leg, left leg, trunk, and total body. We also recorded body mass index (as weight (in kg)/height (in m^2^). Altered clinical conditions impeded assessment of height in 2 patients. Handgrip (dominant hand) was also assessed by a Collins dynamometer to the 115 patients.

In order to assess the presence or not of osteosarcopenic obesity (OSO), we followed recently reported criteria [71,72], defining sarcopenia as a handgrip strength lower than 28 kg (OSO handgrip) and an appendicular skeletal mass index (appendicular skeletal mass (in kg)/height (in m^2^)) lower than 7.26 kg/m^2^ (OSO lean mass); OSO osteoporosis as a femoral neck T-score ≤ 2.5; osteopenia as a femoral T-score ≤ 1; and OSO obesity as percent body fat ≥ 25%.

In order to obtain information about the prevalence of OSO handgrip, we also assessed handgrip strength to 34 hospital workers; 3 (8.82%) fulfilled the OSO handgrip criterion; 26 of them also underwent bone densitometry (not whole body composition); 7.7% were osteoporotic, and 30.8% showed osteopenia.

### 4.2. Laboratory Evaluation

Routine laboratory testing was carried out for all patients, including serum vitamin D levels. Within the first 72 h after admission, blood samples were taken at 8.00 a.m. in fasting conditions and were immediately frozen at −80 °C. The following parameters were determined by Luminex^®^ Performance Assay (R&D Systems, Minneapolis, MN, USA): tumor necrosis factor alpha (TNF-α), with a sensitivity of 0.29 pg/mL; interleukin 8 (IL-8), with a sensitivity of 1.97 pg/mL; and interleukin 6 (IL-6), with a sensitivity of 0.36 pg/mL (data provided by the manufacturer).

### 4.3. Statistics

The Kolmogorov–Smirnov test was used to assess if the variables to be analyzed were normally distributed or not. Nonparametric tests, such as Mann–Whitney’s U test and Kruskal–Wallis test were used to analyze differences or correlations among nonparametric variables. When the variables subjected to analysis showed a normal distribution, Student’s t test and variance analysis were used. We also performed multivariate analyses (stepwise logistic regression and/or multiple correlation analyses) to test the independence or not of the relationships between OSO and several variables. All these analyses were performed with the SPSS program version 25 (Chicago, IL, USA).

## 5. Conclusions and Future Prospects

We conclude that people with alcohol use disorder showed a high prevalence of OSO, especially regarding OSO obesity, OSO osteopenia, and OSO lean mass. OSO handgrip was closely, independently related to the presence of vascular calcification in the present study, a well-known prognostic factor. OSO handgrip was also related to several proinflammatory cytokines (supporting the possible pathogenetic role of these cytokines on OSO development) and vitamin D. Vitamin D deficiency kept, indeed, a close correlation with OSO handgrip, suggesting its pathogenetical involvement in sarcopenia in people with alcohol use disorder. The relationships of OSO handgrip with vascular calcification underscores the need to analyze the impact of both factors on mortality in alcoholic patients. It would be also important to test whether correction of vitamin D deficiency, or implementation of therapeutic measures directed to reduce chronic inflammation in people with severe alcohol use disorder, could improve the OSO scores and, consequentially, the prognosis of these patients.

## Figures and Tables

**Figure 1 ijms-24-09976-f001:**
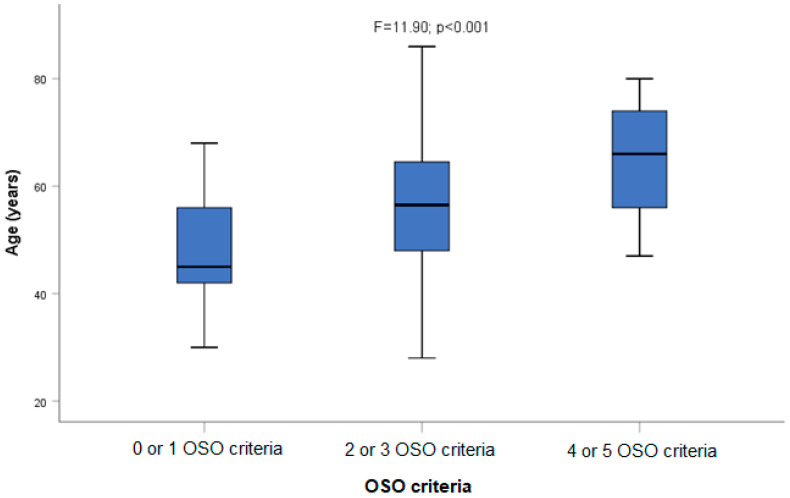
Association between OSO criteria and age.

**Figure 2 ijms-24-09976-f002:**
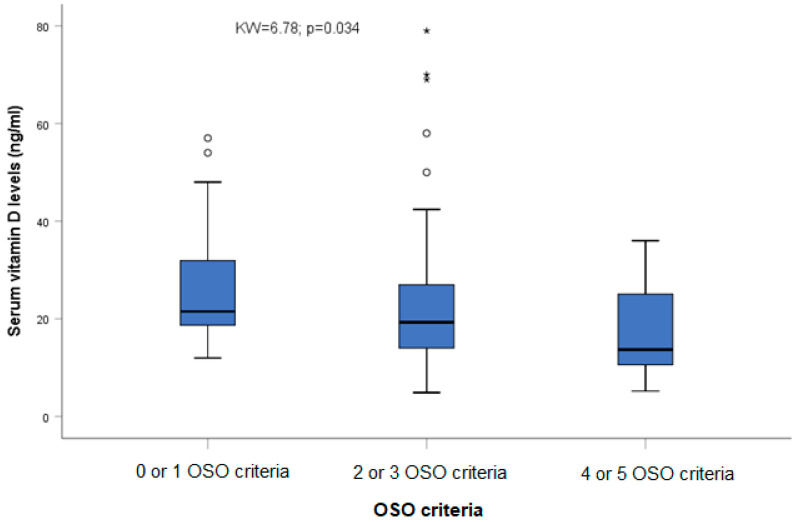
Association among OSO criteria and serum vitamin D levels. Circles correspond to outliers, and asterisks correspond to extreme outliers.

**Figure 3 ijms-24-09976-f003:**
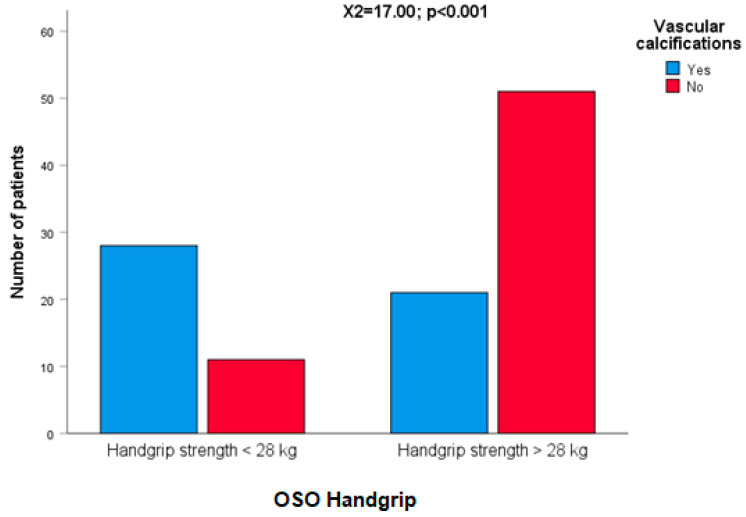
OSO handgrip was significantly associated with vascular calcification.

**Table 1 ijms-24-09976-t001:** Prevalence of each of the OSO criteria among people with cirrhosis and people without cirrhosis.

	Cirrhotics (n = 56)	Non Cirrhotics (n = 59)	
OSO osteopenia	35/56 (62.5%)	29/59 (49.2%)	χ^2^ = 1.57; *p* = 0.21
OSO osteoporosis	15/56 (26.8%)	7/59 (11.9%)	χ^2^ = 3.23; *p* = 0.07
OSO obesity	30/56 (53.6%)	39/59 (66.1%)	χ^2^ = 1.39; *p* = 0.24
OSO handgrip	20/56 (35.7%)	19/59 (32.2%)	χ^2^ = 0.04; *p* = 0.84
OSO lean mass	35/54 (64.8%)	33/59 (55.9%)	χ^2^ = 0.60; *p* = 0.44

OSO: osteosarcopenic obesity.

**Table 2 ijms-24-09976-t002:** Differences in total hip T-score, fat ratio, and handgrip strength were observed when compared to OSO criteria stratified by 0 or 1, 2, or 3 and 4 or 5.

	0 or 1 OSO Criteria (n = 30)	2 or 3 OSO Criteria (n = 68)	4 or 5 OSO Criteria (n = 15)	
Total hip T-score	−0.46 ± 0.95 −0.47 (−0.94–0.02)	−0.89 ± 1.32 −1.03 (−1.75–0.30)	−1.98 ± 0.67 −1.99 (−2.50–1.60)	KW = 21.11; *p* < 0.001
Proportion of fat	23.25 ± 7.23 22.17 (17.66–29.41)	28.95 ± 8.98 29.37 (23.02–36.20)	30.54 ± 8.13 31.60 (25.77–34.46)	KW = 11.46; *p* = 0.003
Handgrip strength	137.33 ± 80.50 140.00 (73.75–190.00)	80.53 ± 73.97 60.00 (16.50–127.50)	42.73 ± 80.84 14.00 (1.00–30.00)	KW = 20.58; *p* < 0.001

OSO: osteosarcopenic obesity.

**Table 3 ijms-24-09976-t003:** Serum cytokines and vitamin D values in patients and healthy individuals.

	IL-6 (pg/mL)	IL-8 (pg/mL)	TNF-α (pg/mL)	Vitamin D (ng/mL)
Patients (n = 115)	5.45 (5.00–15.90)	18.40 (11.38–37.10)	4.30 (1.82–8.70)	20.00 (14–29)
Healthy controls (n = 34)	0.92 (0.69–5.00)	8.95 (5.00–13.02)	5.7 (4.75–8.72)	>30 *
	Z = 6.05; *p* < 0.001	Z = 5.34; *p* < 0.001	Z = 1.23; *p* = 0.22	

* = normal values in our laboratory.

**Table 4 ijms-24-09976-t004:** Relationships between age, proinflammatory cytokines, vitamin D, and handgrip assessed by stepwise multiple regression analyses.

	Unstandardized Coefficients		Standardized Coefficients			95% Confidence Interval for B	
Variable	B	Standard Error	Beta	T	*p*	Lower Bound	Upper Bound
Serum vitamin D	2.489	0.514	0.416	4.838	0.000	1.469	3.508
IL-6	−1.162	0.363	−0.269	3.199	0.002	−1.882	−0.442
Age	−1.314	0.560	−0.201	−2.347	0.021	−2.423	−0.204

Dependent variable: Handgrip. Non-selected variables: TNF-α, IL-8.

**Table 5 ijms-24-09976-t005:** Relationships between age, proinflammatory cytokines, vitamin D, and OSO handgrip assessed by logistic regression analysis.

							95% Confidence Interval for Exp (B)	
Variable	B	Standard Error	Wald	gl	*p*	Exp (B)	Lower Bound	Upper Bound
Serum vitamin D	2.387	0.507	22.124	1	0.000	10.880	4.024	29.416
IL-8	−1.477	0.495	8.901	1	0.003	0.228	0.086	0.602
Age	−1.424	0.523	7.428	1	0.006	0.241	0.086	0.670

Dependent variable: OSO handgrip. Non-selected variables: TNF-α, IL-6.

**Table 6 ijms-24-09976-t006:** Relationships between age, vitamin D, OSO handgrip, and vascular calcifications assessed by logistic regression analysis.

							95% Confidence Interval for Exp (B)	
Variable	B	Standard Error	Wald	gl	*p*	Exp (B)	Lower Bound	Upper Bound
OSO Handgrip	1.822	0.440	17.118	1	0.000	6.182	2.608	14.652
Age	−1.355	0.447	9.212	1	0.002	0.258	0.107	0.619

Dependent variable: Vascular calcifications. Non-selected variables: vitamin D.

## Data Availability

The data presented in this study are available on request from the corresponding author.

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
