# Peer review of "Sarcopenic Obesity in People with Alcoholic Use Disorder: Relation with Inflammation, Vascular Risk Factors and Serum Vitamin D Levels"

_ijms, 2023, doi:10.3390/ijms24129976_

Round 1

Reviewer 1 Report

Specific comments for the authors

Lines 35-62, In the introduction part, Sarcopenic obesity should be mentioned first and it would be more appropriate to talk about the state of this disease in the world. Then the relationship between alcohol and Sarcopenic obesity should be explained. Lines 63-86 should be the first paragraph. Sarcopenic obesity and excessive alcohol consumption, inflammation, vascular heart risks and the mechanisms of serum vitamin D effect should be explained.

Line 307, In which years the samples taken from the patients in the study were taken and in which hospital they were made should be written in this section.

Lines 362-364 “The study protocol was approved by the local ethical committee of our Hospital (number CHUC_2019_83) and conforms to the ethical guidelines of the 1975 Declaration of Helsinki. All the patients gave their written informed consent.” It would be more convenient to move it to line 329.

Lines 234-235, This part should be supported by literature.

Author Response

Reviewer 1.-

Thank you very much for your comments.

Lines 35-62, In the introduction part, Sarcopenic obesity should be mentioned first and it would be more appropriate to talk about the state of this disease in the world. Then the relationship between alcohol and Sarcopenic obesity should be explained. Lines 63-86 should be the first paragraph. Sarcopenic obesity and excessive alcohol consumption, inflammation, vascular heart risks and the mechanisms of serum vitamin D effect should be explained.

We have followed your indications relative to the introduction. Now this section begins mentioning first OSO and commenting some reports on OSO prevalence in the world, and then, we focus on the potential mechanisms involved in the development of OSO in alcoholics.

Line 307, In which years the samples taken from the patients in the study were taken and in which hospital they were made should be written in this section.

Patients were included from 2010 to 2017. Selection criteria are provided. We also describe some characteristics of our hospital, as other reviewers also request.

Lines 362-364 “The study protocol was approved by the local ethical committee of our Hospital (number CHUC_2019_83) and conforms to the ethical guidelines of the 1975 Declaration of Helsinki. All the patients gave their written informed consent.” It would be more convenient to move it to line 329.

We have moved the statement about informed consent, as indicated.

Lines 234-235, This part should be supported by literature.

We have added references to support the statement written in (former) lines 234-235, as requested.

Reviewer 2 Report

Dear authors;

The aim of the present study was to analyze the prevalence of OSA criteria (obesity, handgrip/lean mass, osteopenia/osteoporosis) among alcoholics, the relationship of OSA with some common complications of alcoholics, such as cirrhosis, hypertension, cancer, or vascular calcifications, and with proinflammatory cytokine and vitamin D levels, given the possible pathogenetic role of inflammation on OSA development, and the relevance of the metabolic role of vitamin D both on bone, muscle, and adipose tissue.

This is a well-organized manuscript, the introduction provides enough information and the research design is appropriated. Just moderate editing of English language are needed.

Kind regards, 

Dear authors;

The aim of the present study was to analyze the prevalence of OSA criteria (obesity, handgrip/lean mass, osteopenia/osteoporosis) among alcoholics, the relationship of OSA with some common complications of alcoholics, such as cirrhosis, hypertension, cancer, or vascular calcifications, and with proinflammatory cytokine and vitamin D levels, given the possible pathogenetic role of inflammation on OSA development, and the relevance of the metabolic role of vitamin D both on bone, muscle, and adipose tissue.

This is a well-organized manuscript, the introduction provides enough information and the research design is appropriated. Just moderate editing of English language are needed.

Kind regards, 

Author Response

Reviewer 2.

The aim of the present study was to analyze the prevalence of OSA criteria (obesity, handgrip/lean mass, osteopenia/osteoporosis) among alcoholics, the relationship of OSA with some common complications of alcoholics, such as cirrhosis, hypertension, cancer, or vascular calcifications, and with proinflammatory cytokine and vitamin D levels, given the possible pathogenetic role of inflammation on OSA development, and the relevance of the metabolic role of vitamin D both on bone, muscle, and adipose tissue.

This is a well-organized manuscript, the introduction provides enough information and the research design is appropriated. Just moderate editing of English language are needed.

Response: Thank you very much for your comments. We have tried to correct English, and also followed all the indications provided, in this sense, by other reviewers.

Reviewer 3 Report

Comments

1. The abstract needs a complete modification. The key highlights of the author’s contribution should be mentioned in the abstract. Include numbers with significant p- values in the result section of the abstract.

2. The novelty, rationale, and motivation of this work must be highlighted in the introduction section.

3. The rationale of the current research should be strengthened in the introduction.

4. Future directions should be added to the conclusion.

5. Language editing is required to improve the quality of the manuscript. The author should recheck this manuscript carefully and remove typos and grammatical errors.

6. All references should be thoroughly checked, and especially Author must confirm only relevant publications should be cited.

Specific comments

1.     In the introduction along with normal physiology of nutritional status authors should emphasize it with alcohol-related diseases.

2.     Clearly mention sample size, sample size calculation, and area covering sample size (Village, City, or district).

3.     Study period is not clear, did all patients were analyzed at the same time?

4.     According to the authors, there is a positive correlation between OSA and alcoholism. OSA has also been linked to some of the more prevalent consequences of alcoholism, including diabetes, cancer, hypertension, and vascular calcifications. Despite being a cohort retrospective study, as age also affects OSA, the findings must be compared with those of healthy patients. How do the authors explain the measured parameters in relation to other risk factors? The results need to be explained along with the proper comparative study of non-alcohol risk factors like age, etc.

5.     The conclusion is very broad and weak statements. Specific findings are to be highlighted in future directions.

Language editing is required to improve the quality of the manuscript. The author should recheck this manuscript carefully and remove typos and grammatical errors.

Author Response

Reviewer 3

Thank you very much for your comments. We have tried to respond to all your queries, as commented below.

General comments:

 The abstract needs a complete modification. The key highlights of the author’s contribution should be mentioned in the abstract. Include numbers with significant p- values in the result section of the abstract.

Response: The abstract has been rewritten according to your suggestions, adding p values and numbers in relation to some relevant results, and stressing the relevance of the finding of a relationship between OSO handgrip and vascular calcifications.

  1. The novelty, rationale, and motivation of this work must be highlighted in the introduction section.
  2. The rationale of the current research should be strengthened in the introduction.

Response points 2 - 3. Following your indication and those of other reviewers, the introduction has been fully rewritten and reshaped, underscoring stressing the importance of inflammatory cytokines both in the development of OSO and also explaining that adipokines and other cytokines might play a role in the development of potential consequences associated with OSO.

 Future directions should be added to the conclusion.

Response. We have also reshaped the conclusion, adding some ideas for future research in some aspects of this complex and vast field of knowledge.

 Language editing is required to improve the quality of the manuscript. The author should recheck this manuscript carefully and remove typos and grammatical errors.

Response. We have tried to correct English, and also followed all the indications provided, in this sense, by other reviewers.

All references should be thoroughly checked, and especially Author must confirm only relevant publications should be cited.

Response. We have deleted some references and added several others considered relevant.

Specific comments.

 In the introduction along with normal physiology of nutritional status authors should emphasize it with alcohol-related diseases.

Response. We have commented mechanisms by which ethanol consumption may alter nutritional status.

  1. Clearly mention sample size, sample size calculation, and area covering sample size (Village, City, or district).
  2. Study period is not clear, did all patients were analyzed at the same time?

Response points 2 - 3. Details about included patients are provided. This is a retrospective analysis of prevalence of OSO criteria and relationships of them with clinical complications, proinflammatory cytokines and vascular calcification using prospectively collected data of patients. Out of the ≈ 450 patients included in several prospective studies during the≈ 7-year study period, in this research we included only those patients in whom all the data required for this study were assessed. There is no control group in the strict sense, but, as you and other reviewers wisely suggested, we have provided data of age and sex-matched apparently healthy hospital workers, that were also prospectively collected (as controls) in former studies, although we lack whole body DEXA data of most of these hospital workers.

  According to the authors, there is a positive correlation between OSA and alcoholism. OSA has also been linked to some of the more prevalent consequences of alcoholism, including diabetes, cancer, hypertension, and vascular calcifications. Despite being a cohort retrospective study, as age also affects OSA, the findings must be compared with those of healthy patients. How do the authors explain the measured parameters in relation to other risk factors? The results need to be explained along with the proper comparative study of non-alcohol risk factors like age, etc.

Response. Data about the presence or not of OSO handgrip and OSO osteopenia/osteoporosis of the healthy hospital workers are provided.

  The conclusion is very broad and weak statements. Specific findings are to be highlighted in future directions.

Response.  As commented, we have reshaped the conclusion.

Reviewer 4 Report

In the article entitled " Sarcopenic obesity in alcoholics: relation with inflammation, vascular risk factors and serum vitamin D levels". The authors have investigated the prevalence of some osteosarcopenic obesity criteria in alcoholic patients, highlighting the relationship between this disease and some common complications, proinflammatory cytokine levels, and alteration of vitamin D in alcoholism. However, there are some suggestions to improve the manuscript.

1.       The abstract should be a single paragraph without headings. Please use the International Journal of Molecular Science Instructions for Authors to improve it.

2.       The abbreviation "OSA" refers to osteosarcopenic adiposity, however, the abbreviation "OSO" refers to osteosarcopenic obesity. Please correct it.

3.       p. 1, line 35; Nutritional status is a very important → Nutritional status is an important

4.       p. 1, line 40; com-plex → complex

5.       p. 1, line 41; considered as a structural component → considered structural components

6.       p. 1, line 42; aminoacids → amino acids

7.       p. 2, line 45; fuel reserve → fuel reserves

8.       p. 1, line 41; considered as a physiologic → considered a physiologic

9.       p. 2, line 80; for food → in food

10.    p. 2, line 81; It is therefore → Therefore, it is

11.    p. 2, line 84; since long time → since a long time

12.    p. 2, line 89; complications of alcoholics → complications of alcoholism

13.    p. 2, line 92; vitamin D both on → vitamin D in both the

14.    p. 3, line 96; Prevalence of OSA → The prevalence of OSA

15.    p. 3, line 100; trend to a → trend towards a

16.    p. 3, line 115; OSA criterion consuming → OSA criterion consume

17.    p. 3-4, lines 127, 134, 150; on addiction → of addiction

18.    p. 4, line 146; among OSA and age → between OSA and age 

19.    p. 4, line 147; being significantly → are significantly

20.    p. 4, line 148; In relation with this → In relation to this

21.    p. 4, line 152; among OSA criteria and age → between OSA criteria and age

22.    p. 5, line 172; among IL-8 and handgrip → between IL-8 and handgrip

23.    p. 5, line 173; among IL-6 and handgrip → between IL-6 and handgrip

24.    p. 5, line 174; among handgrip and vitamin D → between handgrip and vitamin D

25.    p. 5, line 174; but relationship → but the relationship

26.    p. 6, line 187; from age → of age

27.    p. 6, line 187; entering in the first place → entering first place

28.    p. 6, line 190; vitamin D being → vitamin D was

29.    p. 7, line 204; even also → even though

30.    p. 8, line 226; among proinflammatory → between proinflammatory

31.    p. 8, line 229; in relation with OSA → in relation to OSA

32.    p. 8, line 232; Secondly, clinical → Secondly, the clinical

33.    p. 8, line 238-239; of them were → of they were

34.    p. 8, line 241; of OSA with age → between OSA and age

35.    p. 8, line 257; with direct → with the direct

36.    p. 8, line 259; may be not associated to → may not be associated with

37.    p. 8, line 260; whichever → whatever

38.    p. 8, line 261; - especially trunk fat- → especially trunk fat

39.    p. 8, line 266; described an → described as an

40.    p. 8, line 274; muscle are → muscles are

41.    p. 9, line 275; effects having been → effects have been

42.    p. 9, line 279; not by → not with; muscle but → muscles but

43.    p. 9, line 297; Although with → Although, there is

44.    p. 9, line 399; when liver → when the liver

45.    p. 9, line 300; contribute to decrease → contribute to decreased

46.    p. 9, line 301; Both more → Both have more

In the manuscript entitled " Sarcopenic obesity in alcoholics: relation with inflammation, vascular risk factors and serum vitamin D levels ". The authors have written the manuscript in comprehensive English with moderate grammatical errors.

Author Response

Reviewer 4

In the article entitled " Sarcopenic obesity in alcoholics: relation with inflammation, vascular risk factors and serum vitamin D levels". The authors have investigated the prevalence of some osteosarcopenic obesity criteria in alcoholic patients, highlighting the relationship between this disease and some common complications, proinflammatory cytokine levels, and alteration of vitamin D in alcoholism. However, there are some suggestions to improve the manuscript.

Thank you very, very much for your detailed corrections, all of them incorporated into the new version of the manuscript, that has been considerably reshaped following the indications of you and those of other reviewers.

Reviewer 5 Report

This manuscript by Martín-González et al details a retrospective analysis of the relationship between alcohol consumption and sarcopenic manifestations.

 The work is well described and the results shown are adequately interpreted, supporting the conclusions drawn by the authors. I have only a couple of issues that I believe the authors can address or explain very easily.

- gender: the sample features 10 women and 195 patients. is there any expected gender dimension in this study? would removing the 10 women change the results at all?

- reference values: are there any reference values for, for example, serum vitamin D levels? from figure 2, there is a clear decrease with the increasing number of OSA criteria. But what are the expected values in non-alcoholic and/or non-OSA positive patients? although the work itself is clear and complete, this relation to the "normal" population could further strengthen the conclusions about OSA- and alcohol-associated dysregulations in the measured variables.

- confiding aspects. IL8 is higher in patients with positive OSA handgrip criterion (line 191), while IL6 is inversely related (line 176). IL6 has been described as both pro- and anti-inflammatory, while IL8 triggers neutrophil chemotaxis. what is the inflammatory overall status of these patients? are there any underlying inflammatory conditions that should/could be encompassed?

Some minor English language errors are present throughout the manuscript, these can be easily corrected and do not all prevent understanding the text. Two of these include:

Line 325 – please change Child score to Child’s score

Table 1, and also other places in the manuscript - what is X2, chi-squared?

Author Response

Reviewer 5

Thank you very much for your encouraging comments.

This manuscript by Martín-González et al details a retrospective analysis of the relationship between alcohol consumption and sarcopenic manifestations.

 The work is well described and the results shown are adequately interpreted, supporting the conclusions drawn by the authors. I have only a couple of issues that I believe the authors can address or explain very easily.

- gender: the sample features 10 women and 195 patients. is there any expected gender dimension in this study? would removing the 10 women change the results at all?

Response. By a couple of complex reasons (an important one, working in a hospital attending many people of rural areas and/or small towns, in which still exists the consideration of drinking alcohol by women as a shameful habit), the proportion of women is always very small (in all our research dealing with alcoholism). This is the reason by which there are only 10 women included. As you suggest, following your advice we have (internally) made the same analyses with only men, and the results are very similar to those obtained with the whole sample. If strictly necessary, we could completely re-analyze our data and rewrite the manuscript including only, but we believe that the proportion of women approximately reflects what we attend in the hospital. 

- reference values: are there any reference values for, for example, serum vitamin D levels? from figure 2, there is a clear decrease with the increasing number of OSA criteria. But what are the expected values in non-alcoholic and/or non-OSA positive patients? although the work itself is clear and complete, this relation to the "normal" population could further strengthen the conclusions about OSA- and alcohol-associated dysregulations in the measured variables.

Response. Also, following your suggestion and those of other reviewers, we have provided data of age and sex-matched apparently healthy hospital workers, that were also prospectively collected (as controls) in former studies, although we lack whole body DEXA data of most of these hospital workers.

- confiding aspects. IL8 is higher in patients with positive OSA handgrip criterion (line 191), while IL6 is inversely related (line 176). IL6 has been described as both pro- and anti-inflammatory, while IL8 triggers neutrophil chemotaxis. what is the inflammatory overall status of these patients? are there any underlying inflammatory conditions that should/could be encompassed?

Response. There is no inconsistency in the results: IL-8 is higher in patients with OSO handgrip criterion (i.e., higher IL-8 values in those with low handgrip values). The same happened with IL-6, and therefore, correlation with handgrip (crude values) is inverse in both cases. The only -in a certain way- surprising results are those related to TNF, but similar results have been reported by other authors, as commented in the discussion. In any case, we have compared our data with those of the “controls”, and both IL-6 and IL-8 are higher in patients, but not TNF, as also some other authors have reported. Cytokines were determined after the acute condition who prompted admission was already controlled, in order to minimize the distortion caused by the acute organic stress on cytokine levels.

Some minor English language errors are present throughout the manuscript, these can be easily corrected and do not all prevent understanding the text. Two of these include:

Line 325 – please change Child score to Child’s score

Table 1, and also other places in the manuscript - what is X2, chi-squared?

We have tried to correct English, and we also followed all the indications provided, in this sense, by other reviewers. We also wrote chi-square correctly.

Round 2

Reviewer 5 Report

Following the author's responses and changes to the manuscript, I believe the manuscript can be published in its present form.